# Environmental Governance, Green Tax and Happiness—An Empirical Study Based on CSS (2019) Data

**Jingjing Wang** [1,2,*]**, Decai Tang** [1,3,*] **and Valentina Boamah** [3]

1   School of Law and Business, Sanjiang University, Nanjing 210012, China
2   Panyapiwat Institute of Management, Bangkok 11120, Thailand
3   School of Management Science and Engineering, Nanjing University of Information Science & Technology, Nanjing 210044, China; 20215242005@nuist.edu.cn
*   Correspondence: wang_jingjing@sju.edu.cn (J.W.); tangdecai@nuist.edu.cn (D.T.)

**Abstract:** The quality of the ecological environment is related to people's health and quality of life, and is a prerequisite for happiness. This paper uses data from the 2019 Chinese Social Survey(CSS) and matches it with green tax data of 30 provinces and autonomous regions in China in 2019 using the mediation effect model to empirically analyze the influence mechanism and internal logic of the environmental governance on happiness. The results show that: (1) environmental governance can significantly improve happiness and indirectly affect happiness through green tax; (2) green tax can significantly enhance happiness; (3) income, regional, and education heterogeneity exists in the direct and mediating effects of environmental governance on happiness. Based on these results, in the context of the new era, we should solidly promote environmental governance and ecological civilization construction, promote the reform of the green tax system, and improve happiness. We should also consider the coordinated development of urban and rural areas and regions and focus on equity and efficiency. In addition, it is necessary to continue to deepen education reform, improve the quality of education, increase people's income, and improve people's happiness. This paper identifies the causal relationship between environmental governance and happiness and helps to clarify the influence mechanism and internal logic of environmental governance on happiness; it also discusses how to address the relationship between governance and development, promote green development, and improve happiness.

**Keywords:** environmental governance; green tax; happiness; mediating effect; heterogeneity

## 1. Introduction

A good ecological environment is the premise and foundation of human survival and development. The economic and social activities of human society are closely related to the ecological environment. Coordinating the relationship between economic development and environmental resources has become the common goal of all countries. Since the implementation of reform and opening-up in 1978, China's economy has continued to grow rapidly for more than 40 years. In 2020, China's GDP exceeded 100 trillion yuan for the first time, ranking second in the world and accounting for about 17% of the world economy [1]. While China's economy is developing rapidly, it also pays the price of wasting resources and the deterioration of the environment. China's environmental pollution, including air pollution, water pollution, land pollution, etc., has become very serious; it causes harm to people's health, and the cost of medical treatment has greatly offset the welfare improvement brought by economic growth, which reduces people's happiness [2,3]. Furthermore, if the problem of environmental degradation cannot be effectively controlled, it will harm the present generation and affect future generations and the ecosystem of the earth. Therefore, environmental governance is an important political issue and a major social issue related to the national economy and people's livelihoods. Environmental

governance cannot be delayed. By managing the environmental pollution problem, the economy can develop sustainably; by managing the environmental pollution problem, people can live in a beautiful environment that will ultimately increase their happiness. At present, the Chinese government has begun to explore effective environmental governance methods, such as continuously increasing investment in environmental protection, setting technical production standards, formulating and improving environmental protection policies, and introducing environmental protection tax and other green tax measures to enhance the effect of the government's environmental governance [4]. With the continuous advancement of environmental governance, will it effectively improve happiness? If environmental governance can significantly improve happiness, what is its mechanism? Are there differences in happiness between different regions, incomes, and educational backgrounds brought by environmental governance? Does green tax play a mediating role between environmental governance and public satisfaction? It is difficult for current research to answer these questions meaningfully. Based on these questions, this paper uses the latest data from the Chinese Social Survey in 2019 and the statistics on green tax to match and conduct in-depth research on the following issues: (1) the relationship between environmental governance, green tax, and happiness; (2) the total and mediating effects of different types of people (such as income level, education, region, etc.) on subjective happiness caused by environmental governance.

## 2. Literature Review

### 2.1. Research on Environmental Governance and Happiness

Environmental pollution, including air, water, and noise pollution, is one of the most important threats to health. According to the World Health Organization, about 25% of diseases worldwide are caused by environmental factors [5]. Many scholars have studied the diseases and harm caused by environmental pollution. They believe that noise pollution, air pollution, etc., affects the immune system, causing cardiovascular diseases, respiratory diseases, and other diseases [6–8]. Ahumada and Victor Iturra (2021) used particulate matter measurements from 305 cities in Chile in 2013 with an instrumental variable strategy and found that air pollution reduces people's happiness [9]. Song et al. (2020) found that the public's subjective perception of air pollution has a significant negative impact on their happiness. The negative effects on happiness of unhealthy people and middle-aged/older people are greater than that of healthy people and young people [10]. In addition, studies have also confirmed regional, education, and income heterogeneity in the impact of environmental pollution on residents' happiness [11]. Since environmental pollution has caused physical and mental illnesses and reduced people's happiness, improving the environment through environmental governance will undoubtedly improve people's happiness. Sanduijav et al. (2021) find that air quality improvement positively relates to happiness [12]. Krekel and MacKerron's (2020) study finds that a green and beautiful environment significantly improves happiness [13].

All of the aforementioned studies believe that environmental governance can decrease environmental pollution and thus improve people's happiness; however, no study uses data from 30 provinces and autonomous regions in China in 2019.

### 2.2. Research on Environmental Governance and Green Tax

Extensive economic development will inevitably damage or pollute the environment. The government is obliged to control environmental pollution and become the main body of environmental governance. It can use administrative, legal, fiscal, and tax means to control the environment. Improving the tax and fee system and implementing a green tax policy have become increasingly important factors to control environmental pollution. Pearce (1991) notes that environmental tax can improve the environment and the efficiency of the tax system, which is the first study to formally establish the idea of double dividends [14]. The study of Pigouvian tax theory is regarded as a precursor to green tax. Bovenberg (1999) believes that green tax is a neutral tax with three potential advantages: improving

environmental quality, reducing tax costs, and increasing labor employment [15]. Gago et al. (2000) believe that the green tax theory is the foundation of modern tax reform. The theory of green tax should include taxes that play a role in fiscal neutrality and a series of policies that the government should issue to regulate environmental pollution [16]. Kwilinski et al. (2019) argue that environmental taxes are an effective environmental policy tool that can control pollution. From the perspective of sustainable development, environmental tax helps enterprises carry out technological innovation [17]. Deng Liping and Chen Bin (2022) studied China's green tax system. Lv et al. (2018) conducted an empirical analysis of the impact of China's green tax policy on the economy [18,19]. The aforementioned literature believes that the green tax is an important means of environmental governance; hence, the research focuses on establishing and improving the green tax system, with environmental tax as the main component, and the effect of policy implementation. Few literature studies can clearly show the causal and quantitative relationship between environmental governance and green tax.

*2.3. Research on Green Tax and Residents' Life Satisfaction*

Green tax can balance the relationship between economic development and the ecological environment. The sustainable development of the social economy and ecological environment can be promoted through green tax. So far, academic research on whether the tax system, tax scale, and macro tax burden affect happiness is relatively scarce. The research conclusions are neither similar or diametrically opposed. Oishi et al. (2012) use a global Gallup poll to show that progressive tax is positively associated with happiness [20]. The research results of Drus M (2016) show that taxes have a positive, significant, and robust effect on happiness [21]. The research of Akay et al. (2012) believes that the impact of tax on happiness is significantly positive [22]. Other studies believe that the state can improve happiness and bring happiness to the public through taxes and increased expenditures on public areas such as education and transportation [23–26].

China's research on tax and happiness conclusions are neither similar or diametrically opposed. The research of Tang Fenglin and Su Lili (2018) points out that, from the overall trend, the happiness of Chinese residents and the growth of the tax scale are positively correlated [27]. The research of Xie Shun et al. (2012) believes that the macro tax burden has a significant negative impact on happiness [28]. The research of Lu Yuanping and Yang Fang (2017) shows that the relationship between China's tax burden and subjective happiness is in an inverted "U"-shaped rising stage, which means that the damage of China's tax burden on residents' happiness is not serious on average. [29]. Zhao Xinyu et al. (2013) believe that there is no certain negative (positive) relationship between the macro tax burden and happiness; that is, the macro tax burden cannot lead to the reduction in happiness [30].

The high-speed growth of China's economy has generally paid the price of sacrificing the environment. Although people's income is increasing, happiness is offset by environmental pollution. Environmental pollution must be controlled; however, there is no consensus on the impact of tax on happiness and economic growth, and there is almost no research on the impact of green tax on happiness. Environmental governance is not only an important part of the government's public governance, but also directly affects happiness; green tax is thus an important means of environmental governance. The research of this paper focuses on happiness from the perspective of environmental governance and green tax, with special emphasis on the impact of environmental governance on happiness and the mediating effect of green tax to enrich the research on happiness. This study also discovers the impact of the green tax on happiness.

The structure of this paper is organized as follows: after introducing the background and literature review, the next section provides the impact mechanism and research hypotheses; from there, the model settings and data sources are described; the metrological tests and results are described next; finally, research conclusions and policy recommendations are given and limitations and future research are discussed simultaneously.

## 3. Influence Mechanism and Research Hypothesis

### 3.1. Analysis of Influence Mechanism

The fundamental purpose of environmental governance is to guide producers to promote environmental protection and greenness, and a green tax is an important means of doing so. When the taxes and fees paid by enterprises are higher than the cost of pollution control, they will naturally choose to increase investment in environmental protection, the interest rate of resource utilization, and reduce environmental pollution; however, it will also force enterprises to carry out technological innovation and upgrades [31,32]. Overall, the levy of green tax mainly plays a role through the price change mechanism, which can guide social behaviour to achieve a "green" transformation. It is also important to promote the protection and conservation of the environment and resources. Environmental governance and green tax are based on green development and driven by reform and innovation, which not only guides enterprises to create a good ecological environment but also improves the level of economic development, promotes employment, and improves happiness and satisfaction to achieve environmental dividends and social benefits, which is called the double bonus [33,34]. The relationship and influence mechanism of environmental governance, green tax, and public happiness are shown in Figure 1.

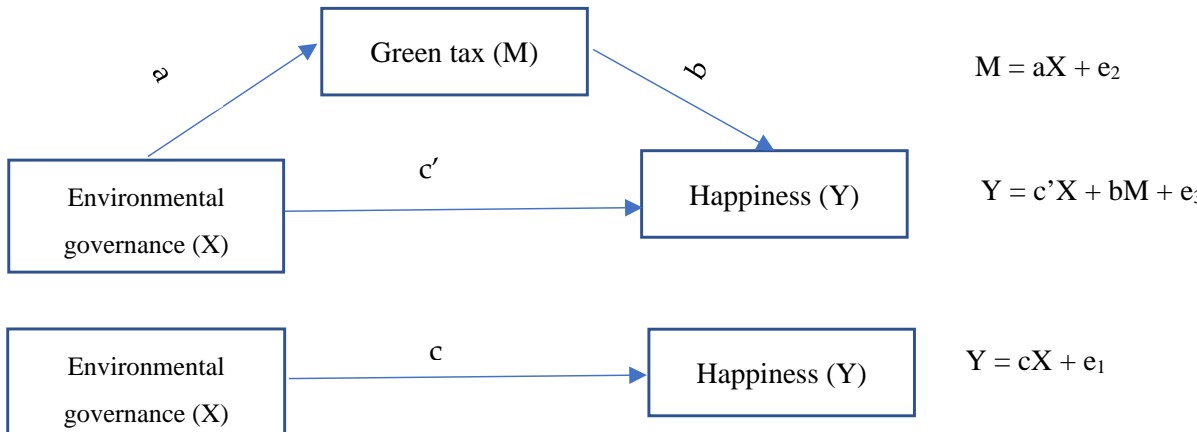

**Figure 1.** The relationship between environmental governance, green tax and happiness. (**a**) The regression coefficient of Environmental governance to Green tax; (**b**) The regression coefficient of Green tax to Happiness; (**c**) The regression coefficient of Environmental governance to Happiness (Total Effect); (**c′**) The regression coefficient of Environmental governance to Happiness (Direct Effect).

### 3.2. Research Hypothesis

In recent years, China's environmental pollution problem has become very serious, and environmental pollutants such as smog and sandstorms are common and must be addressed. Environmental pollution causes huge economic losses to society, inhibits economic growth, and reduces residents' income. Studies have shown that factors such as income, education, age, health, gender, employment, marital status, and social relationships affect happiness [35]. From the perspective of health economics and psychology, environmental pollution will not only cause the human body to suffer from physical diseases such as respiratory, cardiovascular, and cerebrovascular diseases but also lead to mental illnesses such as depression, thereby reducing people's happiness. Therefore, environmental governance will improve the ecological environment and enhance people's happiness. In environmental governance, green tax is an important means, and environmental governance will increase green tax revenue. Although the tax will increase the burden on enterprises, it will also force enterprises to upgrade their industries, implement green innovation, and ultimately achieve the harmonious coexistence and development of enterprises, people and nature, increase social welfare, and enhance people's sense of happiness. Therefore, this paper proposes the following hypothesis:

**Hypothesis (H1).** *Environmental governance can improve happiness.*

**Hypothesis (H2).** *The stronger the environmental governance, the more green tax income it can bring.*

**Hypothesis (H3).** *Green tax can improve happiness.*

**Hypothesis (H4).** *Green tax plays a mediating effect between environmental governance and happiness.*

The problem of unbalanced and insufficient economic and social development in China is still relatively prominent, mainly manifested in unbalanced regional development, uncoordinated urban and rural development, and widening income gaps. As a result, differences among residents are extremely common. The impact of environmental pollution on regions, education degrees, and income groups may differ, so environmental governance on different groups may also be inconsistent. The differences in pollution degree, environmental carrying capacity, and pollution emission intensity between regions, directly determines the impact of environmental governance on happiness; different income/education groups also have different expectations for environmental governance. High-income/education groups prefer green consumption to improve their sense of happiness. The higher the income/education, the higher the preference and requirements for environmental quality. The low-income/education group is more inclined to material consumption to enhance their sense of happiness. [36]. Therefore, the impact of environmental governance on happiness varies according to income and education groups. With this dichotomy in mind, this paper proposes the fifth hypothesis:

**Hypothesis (H5).** *Environmental governance's direct and indirect effects on happiness have heterogeneity in terms of income, region, and education.*

### 4. Model Settings and Data Sources
#### 4.1. Model Settings

This paper constructs a happiness model composed of environmental governance, green tax, and happiness to examine the impact of environmental governance and green tax on happiness by referring to Levinson's econometric model [37]:

$$happ_{ij} = \alpha + \beta 1 envg_{ij} + \beta 2 gret_{j} + \gamma X_{ij} + \varepsilon_{ij} \tag{1}$$

Among them, $happ_{ij}$ is the subjective happiness of residents i in province j, $envg_{ij}$ is the environmental governance intensity subjectively perceived by residents i in province j, $gret_{j}$ is the green tax in province j, $X_{ij}$ is the micro-individual characteristic variable of resident i in province j, and $\varepsilon_{ij}$ is the random disturbance term. The coefficients $\beta 1$ and $\beta 2$ measure the impact of environmental governance and green tax on happiness and the symbols indicate the direction of impact.

#### 4.2. Data Sources

This paper uses 2019 data from the Chinese Social Survey (CSS), a survey conducted by the Chinese Academy of Social Sciences since 2005, an annual survey on the economic status, living conditions, social security and other aspects of Chinese urban and rural households enables a dynamic understanding of social structures, social changes and transformations in all aspects of life. The survey was national, comprehensive, and continuous, which provided advantageous characteristics for our study. The 2019 Chinese Social Survey (CSS) data adopted the random sampling method. More than 11,000 urban and rural households in 30 provinces, 14 cities (counties, districts) and 596 villages were surveyed and 10,283 valid questionnaires were collected, forming 11.6 million data items [38]. Green tax data comes from the China Statistical Yearbook.

*4.3. Variable Selection*

4.3.1. Explained Variables

The explained variable of this paper is happiness, and the data comes from the question in the CSS 2019 questionnaire, "In general, I am a happy person". 1 means strongly agree, 2 means somewhat agree, 3 means disagree, and 4 means strongly disagree. Usually, higher numbers indicate higher happiness, so they are recoded and assigned to a value of 1 for very unhappy, 2 for not very happy, 3 for relatively happy, and 4 for very happy.

4.3.2. Explanatory Variables

The core explanatory variable of this paper is environmental governance, and the data comes from the question, "Do you think the government has done a good job in protecting the environment and controlling pollution?". In the questionnaire, 1 is very good, 2 is relatively good, 3 is not very good, and 4 is very bad. We recoded it and assigned it a value of 1 for very bad, 2 for not very good, 3 for relatively good, and 4 for very good.

4.3.3. Mediating Variables

The mediating variable in this paper is green tax, which uses medium-calibrer data, including environmental protection tax, resource tax, urban land use tax, farmland occupation tax, vehicle and vessel tax, and vehicle purchase tax.

4.3.4. Control Variables

The model in this paper includes the basic control variables involved in most happiness studies: gender, age, age squared, marriage, ethnicity, education level, family income, work status, etc. [39]. In addition, the urban-rural dual structure (agricultural hukou and non-agricultural hukou) and differences in geographical resources (eastern, central, and western) also affect subjective happiness [40,41] and were added as additional variables. The descriptive statistics of the main variables are shown in Table 1.

**Table 1.** Descriptive statistics.

| Variable | | Mean | Std. Dev. | Min | Max |
|---|---|---|---|---|---|
| Happ | 1 = very unhappy; 2 = not very happy; 3 = relatively happy; 4 = very happy | 3.17 | 0.8 | 1 | 4 |
| envg | 1 = very bad; 2 = not very good; 3 = relatively good; 4 = very good | 2.94 | 0.77 | 1 | 4 |
| gret | 15.3–874.19 | 433.38 | 218.15 | 15.3 | 874.19 |
| edu | 1 = junior high school and below; 2 = high school; 3 = Undergraduate; 4 = Graduate and above | 1.57 | 0.81 | 1 | 4 |
| lninc | 0–15.89 | 8.52 | 3.43 | 0 | 14.22 |
| gender | 0 = female; 1 = male | 0.43 | 0.5 | 0 | 1 |
| age | 18–69 years | 46.21 | 14.21 | 18 | 69 |
| age2 | 324–4761 years | 2337.38 | 1274.74 | 324 | 4761 |
| marr | 0 = not married; 1 = married | 0.8 | 0.4 | 0 | 1 |
| minzu | 0 = Minority; 1 = Han | 0.92 | 0.27 | 0 | 1 |
| hukou | 0 = urban; 1 = rural | 0.69 | 0.46 | 0 | 1 |
| work | 0 = non-working state; 1 = working state | 0.65 | 0.48 | 0 | 1 |
| dzx | 1 = East; 2 = Central; 3 = West | 1.86 | 0.81 | 1 | 3 |

To ensure the reliability of the conclusions, this paper partially processes the original data. The 2019 CSS database does not include Xinjiang, Hong Kong, Macau, and Taiwan; therefore, this paper only selects data samples from 30 provincial-level administrative regions. The data of control variables such as gender and age are selected from Part A, with a sample size of 10,286. Happiness is selected from the CAPI random B volume, with a sample size of 5112. Environmental governance is selected from Part G: Social Values and Social Evaluation, with a sample size of 10,286. To maintain data consistency, only 5112

sample data are retained. The sample data for "unclear", "hard to say" and "not suitable" in the questionnaire, as well as the missing values, was deleted in order to obtain 4837 valid sample data.

## 5. Metrological Inspection and Result Analysis

Before the empirical test, to visually present the relationship between environmental governance, green tax and happiness, we draw the fitting curve between the variables, as shown in Figure 2:

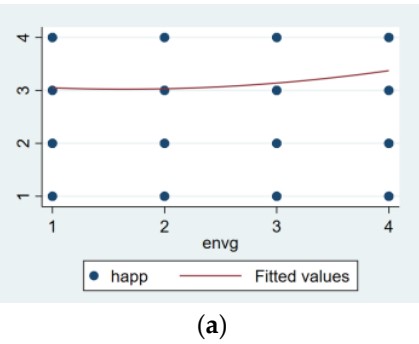

(**a**)

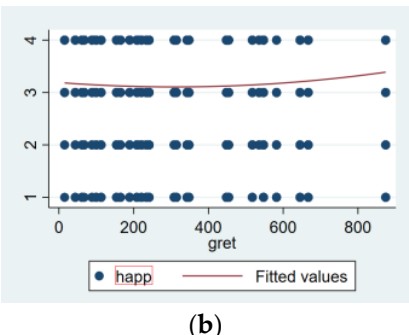

(**b**)

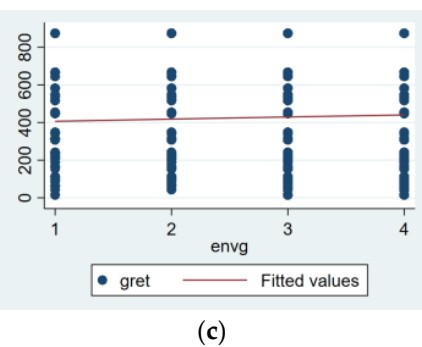

(**c**)

**Figure 2.** Influence relationship between environmental governance, green taxation, and happiness. (**a**) The fitting curve of environmental governance and happiness; (**b**) The fitting curve of green tax and happiness; (**c**) The fitting curve of environmental governance and green tax.

Figure 2a is the fitting curve of environmental governance and happiness. It can be seen that there is a U-shaped curve between environmental governance and people's happiness. Environmental governance can help improve happiness, but the early stage of governance will inhibit the economic development, which is not conducive to improving happiness. With environmental improvement, happiness shows an upward trend. Figure 2b is the fitting curve of green tax and happiness. It can be seen that the relationship between green tax and happiness is a U-shaped curve. The increase in tax expenditure will affect corporate profits and people's income and is not conducive to improving happiness. With the use of green tax for environmental improvement, technological innovation, the transformation and upgrading of enterprises, happiness will rise. Figure 2c shows the fitting curve of environmental governance and green tax, which have a linear relationship: the stricter the environmental governance, the more green tax revenue will be.

### 5.1. Basic Regression Results

Multicollinearity can lead to biased estimation results, so the multicollinearity test is an indispensable step before regression. Multicollinearity is tested using the variance inflation factor (VIF). Age and age squared have multicollinearity, but adding the age squared term is to capture the nonlinear relationship between variables. After removing the age squared variable, the mean VIF is 1.29, and all VIFs are lower than 2, which indicates that the model does not have serious multicollinearity problems [42]. Most of the variables selected in this paper belong to the ordered discrete type, so the Ordered Probit or Ordered Logit model is usually used for estimation; however, in addition to multivariate regression, this paper must study the mediating effect, so the OLS model is used. Many scholars have shown that the signs and significance of Ordered Probit and Ordered Logit or OLS estimation results are consistent [43]. The regression results of environmental governance and green tax on happiness are shown in Table 2.

**Table 2.** Model regression results.

| Variables | (1) Happ | (2) Gret | (3) Happ |
|---|---|---|---|
| envg | 0.149 *** | 15.33 *** | 0.146 *** |
| | (0.0147) | (3.564) | (0.0147) |
| gret | | | 0.000175 *** |
| | | | $(5.93 \times 10^{-5})$ |
| lninc | 0.00913 ** | −1.341 | 0.00937 ** |
| | (0.00372) | (0.902) | (0.00372) |
| edu | 0.0474 *** | 4.196 | 0.0467 ** |
| | (0.0183) | (4.430) | (0.0183) |
| gender | −0.0185 | −11.83 ** | −0.0165 |
| | (0.0237) | (5.755) | (0.0237) |
| age | −0.0484 *** | −4.657 *** | −0.0475 *** |
| | (0.00643) | (1.559) | (0.00643) |
| age2 | 0.000523 *** | 0.0488 *** | 0.000514 *** |
| | $(6.93 \times 10^{-5})$ | (0.0168) | $(6.93 \times 10^{-5})$ |
| marr | 0.283 *** | 26.78 *** | 0.278 *** |
| | (0.0344) | (8.335) | (0.0344) |
| minzu | 0.0534 | 144.1 *** | 0.0282 |
| | (0.0434) | (10.52) | (0.0442) |
| hukou | 0.00489 | 45.22 *** | −0.00304 |
| | (0.0284) | (6.886) | (0.0285) |
| work | −0.0539 * | 11.04 | −0.0559 ** |
| | (0.0281) | (6.813) | (0.0281) |
| dzx | −0.0135 | −107.1 *** | 0.00530 |
| | (0.0145) | (3.514) | (0.0158) |
| Constant | 3.382 *** | 505.7 *** | 3.293 *** |
| | (0.161) | (38.98) | (0.163) |
| Observations | 4837 | 4837 | 4837 |
| R-squared | 0.044 | 0.242 | 0.046 |

Standard errors in parentheses *** $p < 0.01$, ** $p < 0.05$, * $p < 0.1$.

From model (1), after controlling for other variables, there is a significant positive correlation between environmental governance and happiness. For each additional environmental governance unit, happiness increases by 0.149 units; therefore, environmental governance can significantly improve happiness and indicates that Hypothesis 1 is true. Model (2) shows a significant positive correlation between environmental governance and green tax after controlling the other variables. For each additional environmental governance unit, the green tax will increase by 15.33 units, which indicates that hypothesis 2 is true. From model (3), after controlling for other variables, there is a significant positive correlation between green tax and happiness, which indicates that Hypothesis 3 is true. Among the control variables, education, income, age, age squared, marital status, and work are significantly correlated with happiness; gender, ethnicity, hukou, and region are not.

*5.2. The Mediating Relationship between Environmental Governance and Happiness*

From the models (2) and (3) in Table 1, it can be seen that environmental governance has a significant direct effect on happiness; however, the green tax plays a significant mediating effect between the two, which indicates that Hypothesis 4 is true. The KHB method [44] is used to analyze the total, direct, and indirect effects further. The results are shown in Table 3:

**Table 3.** Total effect, direct effect, and indirect effect.

| Variables | (1) happ | % |
|---|---|---|
| Reduced | 0.147 *** | 100 |
| | (0.0147) | |
| Full | 0.144 *** | 98.17 |
| | (0.0147) | |
| Diff | 0.00269 ** | 1.83 |
| | (0.00111) | |
| Observations | 4837 | |

Standard errors in parentheses *** $p < 0.01$, ** $p < 0.05$.

It can be seen from Table 3 that the total, direct, and mediating effects of environmental governance on happiness are all significant. The total effect coefficient of environmental governance on happiness is 0.147, and the direct effect coefficient is 0.144, which accounts for 98.17%. The mediating effect coefficient of green tax on happiness is 0.00269, which accounts for 1.83%. It can be seen that the relationship between environmental governance and happiness is mainly direct, and the mediating effect of the green tax is small but reveals the relationship between green tax and happiness; that is, the green tax does not have a negative effect on happiness but has a significant positive effect. To conclude, the model diagram of the total, direct, and mediating effects of environmental governance on happiness is shown in Figure 3:

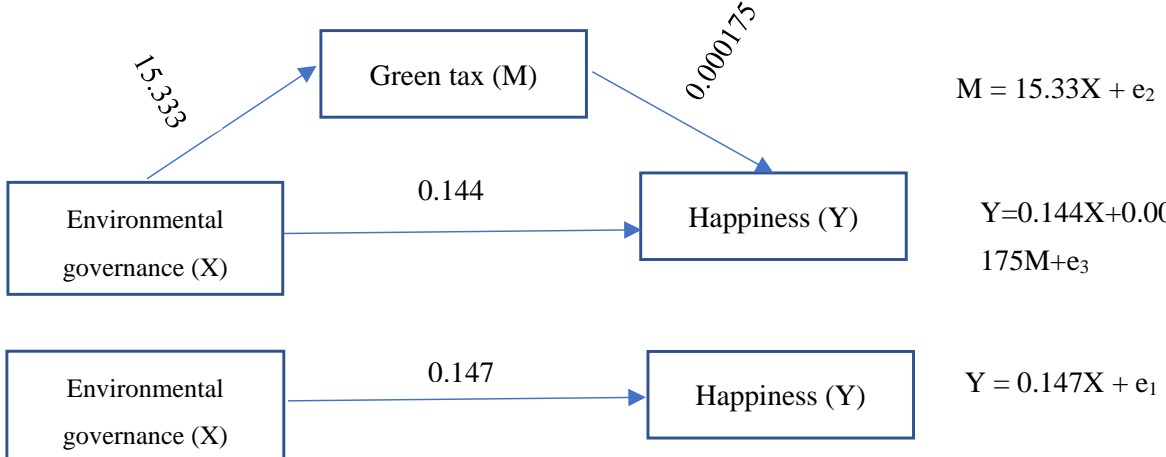

**Figure 3.** The total effect, direct effect, and mediating effect of environmental governance on happiness.

### 5.3. Heterogeneity Research on Environmental Governance

The previous analysis confirmed a mediating effect between environmental governance and happiness. Is there any heterogeneity in this conclusion? Under different incomes, regions, and education levels, what conclusions can be drawn based on the data of CSS2019? We further examine the heterogeneity of the impact mechanism of environmental governance on happiness in the above aspects.

#### 5.3.1. Income Heterogeneity Test

Table 4 shows the mediating effect of happiness in different income groups. In order of individual annual income, those with higher than average annual incomes are classified as the high-income group, and those with lower than average annual income are classified as the low-income group.

**Table 4.** Discussion of income heterogeneity.

| Variables | (1) Hi-Income | % | (2) Low-Income | % |
|---|---|---|---|---|
| Reduced | 0.156 *** | 100 | 0.145 *** | 100 |
| Full | 0.155 *** | 99.39 | 0.141 *** | 97.18 |
| Diff | 0.000950 | 0.61 | 0.00408 ** | 2.82 |
| Observations | 1600 | | 3237 | |

Standard errors in parentheses *** $p < 0.01$, ** $p < 0.05$.

From Table 4, it can be seen that whether it is a high-income or a low-income group, environmental governance still significantly improves residents' happiness, but the degree of impact is different. Its theoretical logic conforms to the environmental Kuznets curve [45]. Specifically, it assumes that happiness is a function of consumer goods and environmental quality. Under the premise of given economic output, residents make trade-offs between consumer goods and a clean environment. For high-income groups, the marginal effect of a clean environment is high. Conversely, the low-income group has a higher marginal effect than a clean environment. Therefore, for high-income groups, a higher-quality environment brought about by environmental governance will improve their happiness. In addition, the high-income group has crossed the income threshold and is not sensitive to the reduction in income caused by green tax; therefore, green tax has not played a significant mediating effect. The reduction in income caused by the green tax will have a greater impact on low-income groups, so the mediating effect of the green tax on low-income groups is significant.

5.3.2. Region Heterogeneity Test

This paper divides China into three regions, east, central, and west, and examines the heterogeneity of the impact of environmental governance on happiness in different regions. The regression results are shown in Table 5.

**Table 5.** Discussion on region heterogeneity.

| Variables | (1) East | % | (2) Central | % | (3) West | % |
|---|---|---|---|---|---|---|
| Reduced | 0.155 *** | 100 | 0.162 *** | 100 | 0.130 *** | 100 |
| Full | 0.150 *** | 96.17 | 0.160 *** | 98.49 | 0.132 *** | 101.31 |
| Diff | 0.00595 ** | 3.83 | 0.00245 | 1.51 | −0.00171 | −1.31 |
| Observations | 1993 | | 1536 | | 1308 | |

Standard errors in parentheses *** $p < 0.01$, ** $p < 0.05$.

Table 5 shows that the total and direct effects of environmental governance on happiness in the eastern, central, and western regions of China are significant. Among them, the total effect is the highest in the central region, followed by the eastern region, and the lowest in the western region. The indirect effect in the east is significant; the indirect effect in the central and western regions is insignificant. In particular, the intermediary effect in the western region is negative because environmental governance has a significant negative effect on green tax. For each additional environmental governance unit, the green tax will decrease by 8.673 units. At the current stage, the economic development of the western region is relatively backward, green tax is relatively scarce, and people pay more attention to increased material wealth. The central and eastern regions have relatively better economic development and higher green tax revenue. People expect to improve the ecological environment and enhance their happiness.

5.3.3. Education Heterogeneity Test

Table 6 shows the mediating effect of happiness in different education groups.

**Table 6.** Discussion on education heterogeneity.

| Variables | (1)<br>happ | % | (2)<br>happ | % |
|---|---|---|---|---|
| Reduced | 0.151 *** | 100 | 0.150 *** | 100 |
| Full | 0.150 *** | 99.84 | 0.147 *** | 97.64 |
| Diff | 0.000236 | 0.16 | 0.00355 ** | 2.36 |
| Observations | 909 | | 3928 | |

Standard errors in parentheses *** $p < 0.01$, ** $p < 0.05$.

Table 6 shows the test results of the impact of different education levels on happiness and environmental governance. In accordance with level of education, people with college and above are divided into a high-education group, and people with high school and below are divided into a low-education group. Regardless of whether it is a high-education or low-education group, the environmental governance's total and direct effects on happiness are significant. The mediating effect of the low-education group is also significant. The happiness brought by environmental governance of the high-education group is slightly higher than that of the low-education group. Usually, higher education will lead to higher income and better careers [46], and members of this group are more concerned about the happiness brought by environmental governance. People with low education are more sensitive to taxes because taxes can reduce income and thus affect happiness.

The above analysis confirms the heterogeneity of the direct effect of environmental governance on happiness and the indirect effect through green tax, which establishes hypothesis 5.

### 5.4. Robustness Check

In general, substituting core variables or using different statistical methods is commonly used for robustness testing. In this paper, the robustness test is conducted by substituting core variables; that is, life satisfaction is used to replace happiness. Table 7 lists the regression results. The results show a significant positive relationship between environmental governance, green tax, and life satisfaction. Table 7 is compared with the results in Table 2. Among the control variables, except for the three variables of ethnicity, hukou, and region, the conclusions of the other variables are consistent; therefore, the research conclusions are robust and good.

**Table 7.** Robustness test estimation results.

| Variables | (1)<br>Satis | (2)<br>Gret | (3)<br>Satis |
|---|---|---|---|
| envg | 0.531 *** | 15.33 *** | 0.516 *** |
| | (0.0397) | (3.564) | (0.0397) |
| gret | | | 0.001000 *** |
| | | | (0.000160) |
| lninc | 0.0432 *** | −1.341 | 0.0446 *** |
| | (0.0101) | (0.902) | (0.0100) |
| edu | 0.300 *** | 4.196 | 0.296 *** |
| | (0.0494) | (4.430) | (0.0492) |
| gender | −0.0804 | −11.83 ** | −0.0686 |
| | (0.0642) | (5.755) | (0.0639) |
| age | −0.146 *** | −4.657 *** | −0.141 *** |
| | (0.0174) | (1.559) | (0.0173) |
| age2 | 0.00161 *** | 0.0488 *** | 0.00156 *** |
| | (0.000187) | (0.0168) | (0.000187) |
| marr | 0.492 *** | 26.78 *** | 0.466 *** |
| | (0.0929) | (8.335) | (0.0927) |

**Table 7.** *Cont.*

| Variables | (1) Satis | (2) Gret | (3) Satis |
|---|---|---|---|
| minzu | 0.0415 | 144.1 *** | −0.103 |
| | (0.117) | (10.52) | (0.119) |
| hukou | −0.108 | 45.22 *** | −0.153 ** |
| | (0.0768) | (6.886) | (0.0768) |
| work | −0.0790 | 11.04 | −0.0901 |
| | (0.0760) | (6.813) | (0.0757) |
| dzx | −0.109 *** | −107.1 *** | −0.00227 |
| | (0.0392) | (3.514) | (0.0426) |
| Constant | 7.674 *** | 505.7 *** | 7.168 *** |
| | (0.435) | (38.98) | (0.440) |
| Observations | 4837 | 4837 | 4837 |
| R-squared | 0.073 | 0.242 | 0.081 |

Standard errors in parentheses *** $p < 0.01$, ** $p < 0.05$.

In addition, after replacing the core explained variables, the mediating effect is shown in Table 8.

**Table 8.** Robustness test of mediation effect.

| Variables | (1) Satis | % |
|---|---|---|
| Reduced | 0.531 *** | 100 |
| | (0.0396) | |
| Full | 0.516 *** | 97.11 |
| | (0.0397) | |
| Diff | 0.0153 *** | 2.89 |
| | (0.00432) | |
| Observations | 4837 | |

Standard errors in parentheses *** $p < 0.01$.

Table 8 shows that after replacing the core variables, the direct and indirect effects of environmental governance on life satisfaction are significant, which is consistent with the conclusions in Table 3, and the model is robust.

## 6. Research Conclusions, Recommendations and Future Prospects

### 6.1. Research Conclusions

The purpose of economic development and social progress is to improve happiness and make people have happier and more satisfying lives; therefore, through environmental governance, we should improve the ecological environment on which people rely so that the sky is bluer, the water is greener, and the air is cleaner. This paper uses the 2019 Chinese Social Survey (CSS) data to match the green tax data and, based on controlling the micro-individual characteristics and variables, conducts an empirical study of the impact and mechanism of environmental governance, green tax, and happiness. Heterogeneity tests are made from three aspects: income, region, and education. The research results show that: (1) improving the environment through environmental governance can promote people's physical and mental health and increase happiness; (2) green tax is an important means of government environmental governance and a green tax can prompt enterprises to increase investment in environmental protection, reduce pollution emissions, improve the environment, and increase happiness; (3) green tax plays a mediating effect between environmental governance and happiness, and the mediating effect has heterogeneity in income, region and education.

The conclusions of this paper remind us that, due to differences in education, income, and regions, people have different requirements for the environment. Environment gov-

ernance policies and tax policies should take these differences into account. Through the rational design of ecological compensation mechanisms and tax rate differences between developed and underdeveloped areas, the income of the poor and low-education groups can be increased to understand the Pareto improvement of social welfare.

This paper identifies the causal relationship between environmental governance and happiness and helps clarify the influence mechanism and internal logic of environmental governance on happiness. It also helps analyze the relationship between governance and development, promote green development, and improve happiness.

### 6.2. Policy Recommendations

At present, countries around the world are striving to manage the environment; under the guise of promoting "carbon peaking and carbon neutrality", examining the impact of environmental governance on happiness has practical policy implications. The details are as follows: (1) The economy should develop, but not at the expense of polluting the environment because is necessary to eliminate outdated production capacity, optimize the economic structure, initiate industrial upgrades, and solidly promote high-quality development. (2) Green tax and other tax methods can significantly improve the environment and enhance happiness, so it is necessary to continuously reform and improve the green tax system with environmental protection tax as the main tax, establish a scientific and standardized green tax system, and give full play to the "double dividend" of green tax. (3) Environmental governance and ecological civilization construction must be strengthened to effectively protect and improve people's livelihood and enhance people's happiness, but the heterogeneity of regions, education, and income must also be considered. For the eastern regions and cities, we should promote clean energy development. It is necessary to develop the economy to increase people's income and improve the ecological environment to enhance people's happiness. Policy support and tax incentives should be increased for the central and western regions and rural areas. Infrastructure construction should be accelerated, regional development should be coordinated, and regional economic development should be promoted to improve happiness. (4) The reform of the income distribution system should be deepened to give full play to the role of tax in the process of income redistribution, adjust excessive income, gradually narrow the income gap, and enhance happiness; deepening the reform of the rural economic system, formulating more policies that benefit farmers, revitalizing the rural economy, and increasing the disposable income of farmers can enhance happiness. (5) The increase of education reform should continue in order to improve the quality of education, allow more people to receive higher and better education, and enhance people's happiness.

### 6.3. Insufficient Research and Future Prospects

This study uses the latest CSS 2019 data for research, which is static cross-sectional data and cannot conduct longitudinal comparison and dynamic research. It is not known whether people's happiness due to environmental governance increased or decreased in 2019 compared to previous years. In the future, multi-year data can be used for transitionary and comparative studies. In the heterogeneity test of environmental governance, we examined the three aspects of income, region, and education. Whether there is heterogeneity in factors such as urban and rural areas, age, and marriage also must be tested. In addition, factors such as fairness, government trust, health, class satisfaction, and social interaction may also mediate environmental governance and happiness. Research on these factors can continue in the future.

**Author Contributions:** Conceptualization, J.W. and D.T.; methodology, J.W; software, J.W.; validation, J.W. and V.B.; formal analysis, J.W.; investigation, J.W.; resources, J.W. and D.T.; data curation, J.W. and D.T.; writing—original draft preparation, J.W.; writing—review and editing, D.T., J.W. and V.B.; visualization, D.T. and V.B.; supervision, D.T.; project administration, D.T. All authors have read and agreed to the published version of the manuscript.

**Funding:** This research received no external funding.

**Institutional Review Board Statement:** Not applicable.

**Informed Consent Statement:** Not applicable.

**Data Availability Statement:** Not applicable.

**Conflicts of Interest:** The authors declare no conflict of interest.

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
