# Peer review of "Environmental Governance, Green Tax and Happiness—An Empirical Study Based on CSS (2019) Data"

_sustainability, doi:10.3390/su14148947_

Round 1
Reviewer 1 Report
This paper uses data from the 2019 China General Social Survey, matches it with green tax, and uses the mediation effect model to empirically analyze the impact of environmental governance and green tax on subjective well-being (happiness). It seems that it is an interesting and attractive topic. The additional ways in which the paper could be improved are:
(1) In the abstract, the research significance needs to be given after the research results.
(2) The research problem of the paper is not clear.
(3) Literature review section, the authors are asked to highlight the findings and limitations of previous studies rather than just listing the purpose of these studies.
(4) Are the conclusions of this paper compared with those of other scholars? Is it consistent with other scholars?
(5) Figures 1-3 should be further explained in more details.
(6) The stated contribution is not clear. The author should highlight the contribution in the conclusion. In addition, the limitations were not well discussed.
(7) The review of the literature could have been much stronger. Such as: Environmental Perceptions, Happiness and Pro-environmental Actions in China; Can Environmental Governance Improve Happiness?-An Empirical Study Based on Cgss Micro Survey Data; Economic benefits of construction waste recycling enterprises under tax incentive policies
Author Response
Dear reviewer, thank you very much for your valuable comments and suggestions. According to your proposal, we have made a substantial revision of the paper from several aspects, such as the abstract, the research problem, literature review and the stated contribution, so as to get your approval.
- Point 1: In the abstract, the research significance needs to be given after the research results.
Response 1: According to your suggestions, we add the research significance. Please see Line 25-28.
- Point 2: The research problem of the paper is not clear.
Response 2: We revise the abstract to make the research problem clear, which is as follows “this paper empirically analyzes the influence mechanism and internal logic of environmental governance on happiness”. Please see Line 11-16.
- Point 3:Literature review section, the authors are asked to highlight the findings and limitations of previous studies rather than just listing the purpose of these studies.
Response 3: According to your suggestions, we add the literature limitations. Please see Line 90-92, 113-116, 143-145.
- Point 4: Are the conclusions of this paper compared with those of other scholars?Is it consistent with other scholars?
Response 4: On the issue that environmental governance affects people's happiness, the conclusions are consistent. However, few papers have studied the mediating effect of green taxation on environmental governance and public happiness. Our study makes a useful complement.
- Point 5: Figures 1-3 should be further explained in more details.
Response 5: According to your suggestions, we further explained Figures1-3.
Please see Line 161-176(Finger1), 288-300(Finger2), 341-346(Finger2)
- Point 6: The stated contribution is not clear. The author should highlight the contribution in the In addition, the limitations were not well discussed.
Response 6: According to your suggestion, we have revised the contribution and limitation. Please see Line 459-463.
- Point 7: The review of the literature could have been much stronger. Such as:Environmental Perceptions, Happiness and Pro-environmental Actions in China; Can Environmental Governance Improve Happiness?-An Empirical Study Based on Cgss Micro Survey Data; Economic benefits of construction waste recycling enterprises under tax incentive policies
Response 7: We have already mentioned the literature of environmental governance and happiness in 2.1. Please see Line 70-92.
Environmental Perceptions, Happiness and Pro-environmental Actions, economic benefits of construction waste recycling enterprises under tax incentive policies are other different topics.
Reviewer 2 Report
The topic of study on environmental governance, green tax and happiness seems to me to be a topic of great interest. And making use of massive data too. The data from the social basket to a large number of people seems interesting to me.
The procedure is suitable
Only one issue that needs to be modified and that is the summary.
The abstract of the article must contain objectives, method (context, participants, data collection instruments, analysis procedures and conclusions). Otherwise, all good
Author Response
Dear reviewer, thank you very much for your valuable comments and suggestions. You suggest that the abstract of the article must contain objectives, method (context, participants, data collection instruments, analysis procedures and conclusions). We think the abstract contains everything except the context and objective, so we add the context and objective to get your approval.
Context: Please see Line 11-13
Objective: Please see Line 25-28
Others factors are shown in abstract as follows:
Participants, data collection instruments analysis procedures: this paper uses data from the 2019 Chinese Social Survey(CSS), matches it with green tax data of 30 provinces and autonomous regions in China in 2019,and uses the mediation effect model…Please see Line 13-16
Conclusions: The results show that: (1) Environmental governance can significantly improve happiness and indirectly affect happiness through green tax. (2) Green tax can significantly enhance happiness. (3) Income, regional, and education heterogeneity exists in the direct and mediating effects of environmental governance on happiness. Please see Line 16-19
Reviewer 3 Report
The study is good and suitable for the journal's theme.
Besides extensive writing, there are a few facts in the introduction section that were not clearly supported by citations. This section should be improved, especially when the authors claimed facts and figures.
For the H3, authors proposed green tax can improved happiness? In the study, the authors mention two groups; people and industry players. Did the hypothesis is concerning both groups? Or a particular group. As we noticed, people will surely be happy, however, did the industry players also happy with this scenario?
The study analyze the mediation effect. How do the authors analyze the mediation effect? Based on which guidelines?
